# Post-Stroke Environmental Enrichment Improves Neurogenesis and Cognitive Function and Reduces the Generation of Aberrant Neurons in the Mouse Hippocampus

**DOI:** 10.3390/cells12040652

**Published:** 2023-02-17

**Authors:** Florus Woitke, Antonia Blank, Anna-Lena Fleischer, Shanshan Zhang, Gina-Marie Lehmann, Julius Broesske, Madlen Haase, Christoph Redecker, Christian W. Schmeer, Silke Keiner

**Affiliations:** 1Department of Neurology, Jena University Hospital, Am Klinikum 1, 07747 Jena, Germany; 2Else Kröner Graduate School for Medical Students, Jena University Hospital, 07747 Jena, Germany; 3Department of Neurology, Lippe General Hospital, Rintelner Strasse 85, 32657 Lemgo, Germany

**Keywords:** MCAO, aberrant neurogenesis, enriched environment

## Abstract

Ischemic lesions stimulate adult neurogenesis in the dentate gyrus, however, this is not associated with better cognitive function. Furthermore, increased neurogenesis is associated with the formation of aberrant neurons. In a previous study, we showed that a running task after a stroke not only increases neurogenesis but also the number of aberrant neurons without improving general performance. Here, we determined whether stimulation in an enriched environment after a lesion could increase neurogenesis and cognitive function without enhancing the number of aberrant neurons. After an ischemic stroke induced by MCAO, animals were transferred to an enriched environment containing a running wheel, tunnels and nest materials. A GFP-retroviral vector was delivered on day 3 post-stroke and a modified water maze test was performed 6 weeks after the lesion. We found that the enriched environment significantly increased the number of new neurons compared with the unstimulated stroke group but not the number of aberrant cells after a lesion. Increased neurogenesis after environmental enrichment was associated with improved cognitive function. Our study showed that early placement in an enriched environment after a stroke lesion markedly increased neurogenesis and flexible learning but not the formation of aberrant neurons, indicating that rehabilitative training, as a combination of running wheel training and enriched environment housing, improved functional and structural outcomes after a stroke.

## 1. Introduction

Stroke is the second major cause of death and a leading cause of permanent disability worldwide [1,2,3]. Stroke mainly results from an occlusion or a severe narrowing of the blood vessels supplying the brain, leading to the death of the associated nerve cells. Patients surviving a stroke often confront permanent physical limitations, and many alterations occurring after a stroke can be explained by the loss of neurons in specific brain regions. Neuronal death following a lesion leads to disturbances in sensory and motor function, balance and speech, and also to cognitive and psychiatric impairments [4,5]. However, changes in cognitive function, including impaired processing speed, memory and orientation, occur independently of the localization of the lesion [6,7]. These forms of impairment are often not recognized in the acute treatment phase, although several studies found some degree of cognitive deficits in up to 70% of patients, depending on the definition and methodology. Moreover, these changes have also been demonstrated in patients after successful clinical rehabilitation [8]. The pathophysiology leading to these changes, which are independent of direct lesions, is still unclear. Currently, stroke treatment is largely limited to reducing ischemic brain damage and rehabilitative training to overcome the associated functional impairment. 

With prompt diagnosis and treatment, a small proportion of patients can be treated with systemic thrombolysis within a time window of three to six hours after the onset of symptoms [9,10]. In addition to this therapy, endovascular thrombectomy has been established in recent years as another treatment for large vessel occlusions [11,12] and has been shown to be superior to thrombolysis alone [13]. Apart from this treatment, which is only feasible in a reduced proportion of patients, there are currently no efficient options to prevent neuronal demise. One therapeutic concept which has been known for a long time is rehabilitative training. It involves promoting independent movement to overcome any paralysis or weakness and often begins in the treating hospital once the condition has stabilized [14]. The physical and occupational therapy treatment concepts are aimed at inducing synaptic remodeling in the brain, which makes it possible to revert some disabilities [15]. Treatments aimed to revert cognitive dysfunction are also conducted in the chronic phase after stroke, but a clear pathophysiological concept does not yet exist [16].

In the adult brain, new neurons are generated from neural stem cells in neurogenic niches in the subventricular zone and the hippocampus via various developmental steps. In the hippocampus [17,18], progenitor cells develop in the subgranular zone (SGZ) of the dentate gyrus, migrate into the granule cell layer and differentiate and integrate into the existing network. Since the hippocampus in the adult brain is responsible for learning processes and spatial orientation, the generation of new neurons taking place here is most likely to play a role in cognitive alterations and the impact of rehabilitation after stroke [19]. The integration of newly formed neurons and their contribution to hippocampal function has been demonstrated in numerous studies [20,21]. 

The process of neurogenesis is influenced by pathologies, such as stroke and other brain lesions, psychiatric diseases or epileptic seizures [19,22,23,24,25]. In particular, after an experimental stroke, there is an increase in neurogenesis, as reflected by enhanced proliferation and differentiation of stem cells, which depends on the size of the infarcts [26,27,28]. This increase in neurogenesis is characterized by an increment in the number of certain progenitor cells in the SGZ of the hippocampus [26], many of which differentiate into mature neurons. The increase in neurogenesis can be further enhanced by interventions, such as rehabilitative training [29,30,31,32,33]. In this context, exercising the paralyzed limb through running wheel training and enriched environment (EE) housing leads to a further increase in the number of newly generated neurons [29,30,32,34]. In addition to the post-ischemic increase in progenitor populations and newly formed neurons, an increased number of morphologically altered aberrant neurons was found [35]. Aberrant neurons have an additional basal dendritic tree or an ectopic location in the hilus. These cells amount up to 1% under physiological conditions and can be as high as 10% after major stroke lesions. Aberrant cells are a pathological feature associated with impaired hippocampal function. Importantly, aberrant neurons become integrated into the existing network [30,35,36,37]; however, their functional significance has not yet been conclusively clarified. It is known that the increased activity in the early stages of neuronal maturation generates abnormal neurons, and these neurons rewire the cortical inputs to the hippocampus. This aberrant neurogenesis is associated with spontaneous seizures [37]. Interestingly, it has been shown that eliminating aberrant neurogenesis reduces spontaneous seizures and normalizes cognitive deficits associated with epilepsy [38]. Abnormal neurogenesis highlights the negative impact that a limited number of aberrant granule cells can have on overall brain function. Understanding aberrant neurogenesis and reducing it after a stroke is essential to preserve and/or improve cognitive function following a lesion.

In our previous study [30], we found that although voluntary wheel running alone increases neurogenesis, it also enhances aberrant neuron formation and worsens cognitive behavior in the Morris water maze task. To date, it is unclear to what extent an EE, consisting of tunnels, nesting material, houses, balls and a running wheel, affects the formation of aberrant neurons and cognitive function after a stroke lesion. In the present study, we found that EE significantly increases neurogenesis after middle cerebral artery occlusion (MCAO) but not the number of aberrant neurons in the dentate gyrus. EE leads to improved flexible learning and higher use of hippocampus-dependent strategies compared with standard (SD) housing. 

## 2. Materials and Methods

### 2.1. Animals and Experimental Design

The study was performed on a total of 72 male C57Bl/6J mice (3 months old). All mice were held in a 14 h light/10 h dark cycle with food pellets and water ad libitum. Mice were randomly divided into 4 groups (MCAO-SD, Sham-SD, MCAO-EE, Sham-EE). The entire evaluation was performed blinded to the experimenter. Mice underwent either an MCAO or sham surgery on day 0 (Figure 1). Postoperatively, mice were kept under standard or EE conditions (6–7 animals/cage, 85 cm × 75 cm × 40 cm) for a total of 2 or 7 weeks (Figure 1). The EE cage was equipped with various objects, such as tube systems, running wheels and houses. The running wheels were freely available to the mice. The arrangement of the objects was changed daily. Animals who survived for 7 weeks were injected intraperitoneally with the thymidine analog, 5-ethynyl-2′-deoxyuridine (EdU, 50 mg/kg) once daily from day 3 to day 15 after stroke, to label proliferating cells. On day 4, all animals underwent stereotactic injections of a retroviral vector into the dentate gyrus. Cognitive function was assessed using the Morris water maze (MWM) test for 5 days, 1 week before perfusion. Mice were transcardially perfused 7 weeks after infarct induction (Figure 1). All procedures were approved by the German Animal Care and Use Committee in accordance with European Directives.

### 2.2. Cerebral Ischemia Model

Ischemic infarcts were induced using the MCAO model. All animals were anesthetized with 2.5% isoflurane and a 3:1 nitrous oxide/oxygen mixture. A midline neck incision was performed to expose the right common carotid artery (CCA). The two bifurcations of the CCA, the external carotid artery (ECA) and the internal carotid artery (ICA), were freed from surrounding tissue, and the CCA and ECA were occluded with a 7.0 polyfilament (Medicon eG, Tuttlingen, Germany). A 6.0 monofilament suture (Doccol Corporation, Sharon, MA, USA) with a rounded tip was inserted into the ICA. The occlusion time was 45 min and resulted in the occlusion of the middle cerebral artery. The suture was then removed, and the wound closed. During MCAO, the body temperature of the mice was maintained with a heating pad. Sham-operated control mice underwent the same surgical procedure. The sham animals underwent the same surgical procedures as the MCAO mice except for the suture occlusion.

The evaluation of the neurological deficits after the induction of the MCAO was conducted with the help of the Bederson score [39,40]. 

### 2.3. Injections of 5-ethynyl-2′-deoxyuridine (EdU)

To label proliferating cells, mice were injected with 5-ethynyl-2′-deoxyuridine (EdU; 50 mg/kg, i.p.; Merck, Millipore, Darmstadt, Germany) once daily for 13 consecutive days, 7 weeks prior to perfusion (Figure 1).

### 2.4. Stereotactic Injections of a Retroviral Vector

For morphological analyses and detection of aberrant neurons, a green fluorescent protein GFP-retroviral vector was injected into the dentate gyrus of all animals on day 4 after surgery. CAG-green fluorescent protein retroviral vectors were developed from a mouse Moloney leukemia virus by co-transfection of HEK 293 T cells with the compound promoter CAG, the reporter gene GFP, the CMV enhancer protein, the VSV-G rabies virus coating glycoprotein and the Woodchuck hepatitis virus post-transcriptional regulatory element (WPRE). The final vector titer was 1 × 10^7^ colony-forming units/mL. Mice were anesthetized with 2–3% isoflurane in a 70% nitrous oxide and 30% oxygen mixture. The following Bregma coordinates were used: AP −2.3 mm, ML ±1.5 mm and DV −2.0 mm. 1.2 µL of the viral vector was injected into the brain tissue ipsilateral to the infarct. During the injection, body temperature was maintained with a heating pad. Sham-operated control mice were subjected to the same surgical procedure.

### 2.5. Tissue Preparation 

On day 14 or 46, mice were sacrificed with an overdose of isoflurane, and transcardial perfusion was performed through the ascending aorta with 4% paraformaldehyde dissolved in phosphate buffer (0.15 mol/L, pH 7.4). The brains were then postfixed in paraformaldehyde, followed by 24 h in 10% sucrose solution and at least 24 h in 30% sucrose solution. The brains were frozen in 2-methylbutane and cut into 40 µm thick slices using a cryomicrotome. 

### 2.6. Stainings

#### 2.6.1. MAP2, MCM2 and DCX Staining 

Immunohistochemical stainings were performed using antibodies raised against minichromosome maintenance protein 2 (MCM2) to quantify endogenous proliferation, microtubulin-associated protein 2 (MAP2) to determine brain, hippocampal and infarct volume and doublecortin (DCX) to analyze precursor cells. The neuronal marker MAP2 can only be detected in undamaged brain tissue [41]. Every 12th brain section was used for volume determination and every 6th section for proliferation and DCX^+^ cell analysis. In the first step, brain sections were washed 6 times for 15 min each in Tris-buffered saline (TBS) buffer solution, followed by incubation with hydrogen peroxide (H_2_O_2_) for 30 min to consume the peroxidases present in the tissue. After washing again 3 times for 15 min in TBS, sections were pre-incubated for 2 h in a solution of TBS Plus (TBS mixed with 0.1% Triton X-100 and 0.3% normal donkey serum), 0.2% bovine serum albumin, 0.3% milk powder and the antigen-binding fragment (Fab, 1:200, Dianova, Hamburg, Germany). Sections were washed in TBS 3 times for 15 min each, followed by incubation with the primary monoclonal antibody mouse anti-MAP2 (1:10,000, Sigma-Aldrich, St. Louis, MO, USA), rabbit anti-MCM2 (1:500, cell signaling, Leiden, The Netherlands) or goat anti-DCX (1:500, Santa Cruz, Dallas, TX, USA) in TBS Plus overnight at 4 °C. On the second day, sections were washed 3 times for 15 min each in TBS and then incubated for 30 min in TBS plus. In the next step, the tissue was transferred to an incubation solution consisting of the secondary antibody biotinylated donkey anti-mouse IgG (1:500, Jackson Immuno Research, Cambridgeshire, UK) or anti-goat IgG (1:500, Dianova, Hamburg, Deutschland) or anti-rabbit IgG (1:500, Dianova, Hamburg, Deutschland) in TBS plus for 2 h. This was followed by 3 washes in TBS for 15 min each and by incubation in avidin-biotin reagent (Vector Laboratories, Burlingame, CA, USA) for 1 h. After washing 3 times in TBS for 15 min each, 0.03% H_2_O_2_ was added to the DAB solution (0.05 mg/ml TBS), and sections were incubated for 8 min. Finally, sections were washed in TBS 3 times. After completion of the staining, sections were mounted on slides with 0.5% gelatin. After drying, slides were covered with Entellan^®^ (Merck Millipore, Darmstadt, Germany).

#### 2.6.2. DCX/MCM2/DAPI Staining 

Precursor cells were identified, and their endogenous proliferation was quantified immunocytochemically by double (7-week groups) or triple-labeling (2-week groups) of every 24th hippocampal section from each brain. After washing three times with TBS and blocked in TBS-plus solution for 30 min, the slices were incubated with primary antibody guinea pig anti-DCX (1:500, Merck Millipore, Darmstadt, Germany) or goat anti-DCX (1:500, Santa Cruz, Dallas, TX, USA) and rabbit anti-MCM2 (Cell signaling, Leiden, The Netherlands) in TBS-plus overnight at 4 °C. The next day the slices were washed with TBS three times, followed by blocking with TBS-plus. After incubation with secondary antibody Alexa488 anti-guinea pig (1:500, MoBiTec, Goettingen, Deutschland), Alexa488 anti-goat (1:500, MoBiTec, Goettingen, Deutschland) and rhodamine anti-rabbit (1:500, Dianova, Hamburg, Deutschland) for 2 h at room temperature, the slices were rinsed in TBS and stained with DAPI solution in a cuvette for 5 min, washed in phosphate-buffered saline and mounted with Moviol. 

#### 2.6.3. EdU/NeuN/DAPI Staining 

For the detection of newly formed neurons, the proliferation marker EdU was used together with the neuronal marker NeuN. For EdU staining, the special Click-iT^®^ chemistry was used. In addition, for immunofluorescence imaging of all cells, nucleus staining with 4′,6-diamidino-2-phenylindole dihydrochloride (DAPI, Sigma–Aldrich, St. Louis, MO, USA) was performed. For staining, the Click-iT^®^ Imaging Kit protocol (Invitrogen, Carlsbad, CA, USA) was first conducted at room temperature. For this, the tissue was first washed twice for 15 min in a solution of PBS with 3% BSA. Next, the sections were incubated for 20 min in PBS containing 0.5% Triton-X100 and then washed again 2 times, analogous to the first step. The Click-iT^®^ reaction mixture was prepared so that it could immediately be used after the last washing step. Sections were then incubated for 30 min. In the end, the sections were washed again twice for 15 min in the BSA-PBS solution. Further, sections were washed 3 times for 15 min each in TBS and then pre-incubated in TBS Plus for 30 min. This was followed by incubation with the primary mouse antibody anti-NeuN (1:500, Chemicon, Temecula, CA, USA) in TBS Plus for 2 days at 4 °C. On the third day, the sections were washed for 30 min each in TBS Plus. This was followed by incubation with the secondary antibody Cy5 anti-mouse (1:500, Dianova, Hamburg, Germany) in TBS Plus overnight at 4 °C. Finally, sections were washed 6 more times in TBS for 10 min each and mounted on slides using 0.5% gelatin. DAPI staining was performed in glass cuvettes for 5 min in DAPI solution and then 3 times in TBS and rinsed with distilled water. The slides were covered with Moviol^®^. All steps involving the Click-iT^®^ reaction mixture were conducted in darkness. 

#### 2.6.4. GFP/NeuN/DAPI Staining 

For the examination of neuronal morphology using GFP labeling, a triple immunofluorescence staining was performed on every 3rd hippocampal slice. Briefly, sections were washed 6 times in TBS for 15 min. This was followed by a 30-min incubation in TBS Plus and subsequent exposure to the primary antibodies mouse anti-NeuN and goat anti-GFP for 2 days at 4 °C. On day 3, sections were first washed again 3 times for 10 min each in TBS and then pre-incubated for 30 min in TBS plus. Sections were then incubated with the secondary antibodies Cy5 anti-mouse and Alexa Fluor 488 anti-goat or rhodamine anti-rat overnight at 4 °C. Analogous to the EdU/NeuN/DAPI staining, sections were mounted on slides after washing 6 times for 10 min each in TBS, DAPI stained and coverslipped with Moviol^®^. The primary antibodies used were goat anti-GFP (1:500; Acris, Herford, Germany), rat anti-RFP (1:500; Abcam, Cambridge, UK), mouse antineuronal nuclei antigen (1:500; Chemicon, Temecula, CA, USA) and Cy5 anti-mouse (1:500; Dianova, Hamburg, Germany). After washing and blocking, the following secondary antibodies were used: Alexa Fluor 488 anti-goat (1:500; Invitrogen, Carlsbad, CA, USA) and Rhodamine anti-rat (1:500; Dianova, Hamburg, Germany). To label the cell nuclei, all sections were additionally stained with DAPI.

### 2.7. Brain and Infarct Volume Quantifications

MAP2-stained sections were used to determine brain, hippocampal or infarct volume. Every 12th brain section was placed on a light table (Northern Light Precision Illuminator, MCID, Washington, DC, USA) and imaged with a digital camera (Hamamatsu Photonics K.K., Wexford Bayne, PA, USA) using Simple PCI software (version 6, Hamamatsu Photonics K.K., Wexford Bayne, PA, USA). Subsequently, brain and infarct areas were determined using Scion Image software (version 4, Scion Corporation, Torrance, CA, USA). The section areas from each brain were added and multiplied by the slice thickness (0.04 mm) and the distance (every 12th section).

### 2.8. Cell Quantification 

Peroxidase-stained MCM2^+^ cells were counted in every 6th section of the complete ipsi- and contralateral dentate gyrus. The total number of MCM2^+^ cells was determined by multiplying the counted MCM2^+^ cells in the dentate gyrus by 6.

The phenotypes of about 100 MCM2^+^ cells were determined by colocalization with DCX, using confocal laser scanning microscopy (LSM 900; Zeiss Jena, Germany). Colocalization was confirmed by z-series through the cell soma allowing for the assessment of overlap between the antigens. The ratio of DCX/MCM2 phenotype was calculated per animal based on the absolute number of MCM2^+^ cells (peroxidase method).

In order to quantify the total number of DCX^+^ precursor cells in the whole dentate gyrus, DCX^+^ cells in every 6th section ipsi- and contralaterally were assessed with a 40× objective. The total number of DCX^+^ cells was determined by multiplying all counted DCX^+^ cells in the dentate gyrus by 6.

In order to quantify neurogenesis in the dentate gyrus, EdU/NeuN/DAPI cells in every sixth section ipsi- and contralaterally were assessed using immunofluorescence staining and confocal microscopy (LSM 710, Carl Zeiss Jena, Germany) with a 40× objective. New neurons were identified as EdU^+^/NeuN^+^ cells. The number of newly formed neurons was determined by calculating the ratio of EdU^+^/NeuN^+^ cells compared with all EdU^+^ cells in the dentate gyrus. The total number of new neurons was determined by multiplying all counted NeuN/EdU^+^ cells in the dentate gyrus by 6. All quantifications were conducted in the subgranular zone and granular cell layer in both hemispheres.

For quantification of aberrant neurons, confocal microscopy z-stack imaging and the semi-automated Neurolucida system (MicroBrightfield, Colchester, VT, USA) were used to analyze the phenotype and dendritic complexity of GFP/NeuN/DAPI stained cells, as previously reported [35]. Neurons with aberrant morphology were defined as bipolar neurons in the subgranular zone or granular cell layer with basal dendritic projections oriented towards the hilus or neurons with an abnormal position of their soma inside the hilus.

### 2.9. Assessment of Neuronal Morphology 

GFP/NeuN/DAPI stained sections were used to quantify the number, length and complexity of dendrites. 3D images from neurons were obtained with a confocal LSM (LSM 710 from Carl Zeiss, Jena, Germany) at 40× magnification using an oil immersion objective. The GFP/NeuN/DAPI positive neurons were divided into typical and aberrant neurons. Typical neurons were characterized by their apical dendritic tree towards the molecular layer and a basal axon towards the hilus. In contrast, aberrant neurons had an additional basal dendrite tree or were characterized by their ectopic position in the hilus. The 3D reconstruction of the dendritic trees was performed using the Neurolucida Neuron Tracing software (MBF Bioscience, Williston, VT, USA). The reconstructed neurons were evaluated with the software NeuroExplorer (MBF Bioscience, Williston, VT, USA). The total length of the dendritic trees, as well as their complexity assessed with a Scholl analysis, were determined. 

### 2.10. Morris Water Maze Test 

For the analysis of the Morris water maze task, quantification of latencies (defined as the time to find the platform), distance (defined as path lengths) and time taken by the animals to find the hidden platform was evaluated. In addition to the classical parameters, hippocampus-dependent and independent search strategies were analyzed. The water maze was conducted over 5 days, 6 weeks post-stroke. Mice had to find a hidden platform in a 1.80 m diameter water-filled pool. The water was clouded with milk powder and set to about 20–21 °C. Each mouse completed 6 trials per day for 5 consecutive days. Between each trial, mice were given a 30-min break. Mice were placed in the pool at one of four possible starting points, which were maintained each day. In total, the animals had 120 s to find the hidden platform in the pool. If an animal did not reach the platform within the time, it was led to the platform and rested there for 15 s. On days 1–3, the platform was in the same place, and on days 4 and 5, it was moved to the opposite quadrant. Probe trials took place on days 4 (before reversal) and 6. All swim paths were recorded with VideoMod2 (TSE version 6.04; Rostock, Germany) and subsequently analyzed with the software Matlab (Mathworks, Apple Hill Drive 1, Natick, MA, USA).

A basic distinction was made between hippocampus-independent (HI) and hippocampus-dependent (HD) strategies. Within these main groups, further developmental steps can be defined. According to specific parameters, the HI strategies are thus divided into Thigmotaxis, Random Search, Scanning and Chaining, and the HD strategies are divided into Direct Search, Focal Search and Direct Swimming. The use of specific hippocampus-dependent and -independent search strategies allowed a detailed analysis of the animals’ behavior. The analysis of the data was conducted with the help of the software Matlab (The Mathworks, Natick, MA, USA). The basis for calculating the search strategies was the division of the maze into X and Y coordinates. With the help of these coordinates, it was possible to calculate the exact path of the mice until they reached the hidden platform.

In addition, Perseverance, defined as a further strategy that develops when the relearning ability is disturbed after the platform has changed position, was assessed. The different strategies in daily sessions were displayed graphically via Matlab software. In addition, the percentage of HD strategies on the respective test day was graphically represented.

### 2.11. Statistical Analysis

Cell quantification, volumetry analyses and probe trails were statistically evaluated using a Mann-Whitney U test. The median (Mdn) and interquartile range (IQR) are given for each group. The quantification of DCX^+^ precursors was analyzed using the ANOVA Test with post-hoc Bonferroni. Data are presented as mean ± SEM. In this exploratory analysis, each *p*-value should be considered as a level of evidence against each null hypothesis. Results from the Sholl analysis were tested using a one-way ANOVA (dependent variable: crosses, factor: group; post-hoc Bonferroni), and from the dendritic length were tested using a linear mixed model (dependent variable: length, factor: group; post-hoc Bonferroni). Data are presented as mean ± SEM. 

For the Morris water maze evaluations, the following statistical analyses were performed: differences in latency, distance or speed between groups were determined using a 2-way ANOVA with repeated measures and post-hoc Bonferroni test (dependent variable: latency, distance or speed; within-subject variables: days and trails; between-subject factor: intervention). The analysis of latency, distance or speed between control and MCAO groups was performed using a 2-way ANOVA with repeated measures and post-hoc Bonferroni test (dependent variable: latency, distance or speed; within-subject variables: days and trials; between-subject factor: groups). For comparison of latency, distance or speed in both groups (MCAO vs. controls) on each day follow-up, a 2-way ANOVA with repeated measures and post-hoc Bonferroni test were used (dependent variable: latency, distance or speed; within-subject variables: Trails; between-subjects factor: groups). 

Hippocampus-dependent strategies in the MCAO and control groups on the different days were analyzed using binary logistic regression and a post-hoc Bonferroni test (dependent variable: hippocampus-dependent strategies; subject variables: animal; between-subjects interaction factor: days, groups or interaction between days and groups). The hippocampus-dependent strategies were evaluated using binary logistic regression (dependent variable: hippocampus-dependent strategies; subject variables: animal; between-subjects factor: groups). 

The different hippocampus-dependent and -independent search strategies used in the MWM were analyzed in an exploratory data analysis using an algorithm based on the method of generalized estimating equations [14]. The classical parameters of the MWM (latency, distance and speed) and the hippocampus-dependent strategies are reported as mean ± SEM. Statistical analyses were performed using SPSS 27.0 for Windows (IBM Corp., Armonk, NY, USA). A *p*-value of <0.05 was considered statistically significant.

## 3. Results

### 3.1. Brain- and Hippocampal Volume

No significant changes were detected in the brain (sham-SD Mdn = 215 mm^3^; IqR = 11 vs. MCAO-SD Mdn = 221 mm^3^; IqR = 27.5; U = 19; *n* = 12; *p* = 0.943; sham-EE Mdn = 234 mm^3^; IqR = 17 vs. MCAO-EE Mdn = 211 mm^3^; IqR = 21.5; U = 12; *n* = 12; *p* = 0.432; sham-EE vs. sham-SD; U = 15; *n* = 15; *p* = 0.152; MCAO-EE vs. MCAO-SD; U = 11; *n* = 10; *p* = 0.841) and hippocampal volumes between sham and MCAO at different housing conditions (hippocampus volume: ipsi: sham-SD Mdn = 9.5 mm^3^; IqR = 2.75 vs. MCAO-SD Mdn = 10 mm^3^; IqR = 3; U = 15; *n* = 13; *p* = 0.524; sham-EE Mdn = 10 mm^3^; IqR = 2 vs. MCAO-EE Mdn = 10 mm^3^; IqR = 53; U = 9.5; *n* = 12; *p* = 0.202; contra: sham-SD Mdn = 10 mm^3^; IqR = 0.75 vs. MCAO-SD Mdn = 10 mm^3^; IqR = 1.5; U = 18; *n* = 13; *p* = 0.833 sham-EE Mdn = 10 mm^3^; IqR = 2 vs. MCAO-EE Mdn = 10 mm^3^; IqR = 2.5; U = 12.5; *n* = 12; *p* = 0.432) 2 weeks post-surgery (Figure 2A,B). 

There were significant changes in the brain volume and hippocampal volume of animals in the standard cage after a stroke (Figure 2). In particular, stroke animals had a lower brain volume compared with the controls (sham-SD Mdn = 227 mm^3^; IqR = 12.74; MCAO-SD Mdn = 208 mm^3^; IqR = 8.86; U < 0.001; *n* = 13; *p* = 0.003) (Figure 2B). This significant decrease was also shown in the stroke animals ipsi- and contralateral to the infarct (ipsi: sham-SD Mdn = 9.48 mm^3^; IqR = 0.29; MCAO-SD Mdn = 8.99 mm^3^; IqR = 1.31; U = 5.00; *n* = 13; *p* = 0.022; contra: sham-SD Mdn = 9.55 mm^3^; IqR = 0.65; MCAO-SD Mdn = 9.17 mm^3^; IqR = 0.71; U = 4.00; *n* = 13; *p* = 0.015). In contrast to the animals in the standard housing, animals in the EE showed no significant differences in either brain volume (sham-EE Mdn = 224 mm^3^; IqR = 9.25; MCAO-EE Mdn = 216 mm^3^; IqR = 40.64; U = 17.00; *n* = 14; *p* = 0.338) or hippocampus volume between ipsi- and contralateral sides (ipsi: sham-EE Mdn = 9.48 mm^3^; IqR = 1.03; MCAO-EE Mdn = 8.87 mm^3^; IqR = 2.02; U = 16.00; *n* = 14; *p* = 0.277; contra: sham-EE Mdn = 9.84 mm^3^; IqR = 0.5; MCAO-EE Mdn = 9.56 mm^3^; IqR = 2.33; U = 22.00; *n* = 14; *p* = 0.749) (Figure 2A,B). 

### 3.2. Time-Dependent Reduction in Infarct Volume under Standard Housing 

All stroke animals included in the study showed typical infarcts in the striatum area. There were no significant differences in the size of the infarct volume between standard and EE housing conditions 2 weeks (MCAO-EE Mdn = 12 mm^3^; IqR = 12; U = 11.5; *n* = 10; MCAO-SD Mdn = 12 mm^3^; IqR = 5; *p* = 0.841) and 7 weeks (MCAO-EE Mdn = 8.55 mm^3^; IqR = 4.99, U = 7.00; *n* = 13; MCAO-SD Mdn = 5.25 mm^3^; IqR = 1.35; *p* = 0.051) post-stroke (Figure 2C). The comparison of the infarct volume between 2 and 7 weeks results in a significant change between MCAO-SD (2 weeks: MCAO-SD Mdn = 12 mm^3^; IqR = 5 vs. 7 weeks: MCAO-SD Mdn = 5.25 mm^3^; IqR = 1.35; U < 0.001; *n* = 11; *p* = 0.004; 2 weeks: MCAO-EE Mdn = 12 mm^3^; IqR = 12 vs. 7 weeks: MCAO-EE Mdn = 8.55 mm^3^; IqR = 4.99; U = 15; *n* = 12; *p* = 0.755) (Figure 2C).

**Figure 2 cells-12-00652-f002:**
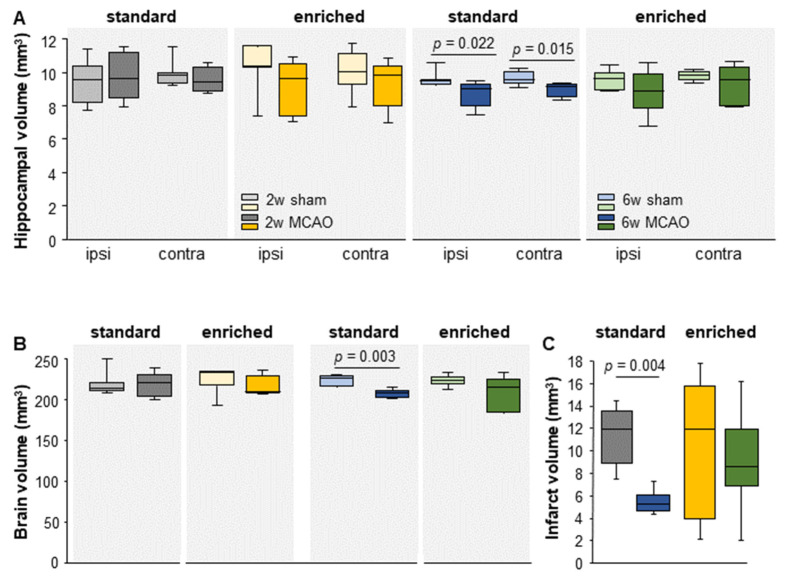
Impact of MCAO and EE on (**A**) hippocampal volume, (**B**) brain volume and (**C**) infarct volume. Significant differences were found in the hippocampal volume between sham and MCAO under standard conditions, both ipsi- and contralateral, 7 weeks post-surgery. The boxplots show the median, upper and lower quartile, and minimum and maximum values. Statistical analyses were performed using the Mann–Whitney U test.

### 3.3. Cell Proliferation after Stroke

Two weeks after stroke induction, significant changes in the proliferation behavior of the MCM2^+^ cells were observed after standard housing. Keeping animals in an EE does not significantly affect proliferation (sham-SD Mdn = 15.240 MCM2^+^ cells; IqR = 3150 vs. MCAO-SD Mdn = 18.528 MCM2^+^ cells; IqR = 6768; U = 3; *n* = 10; *p* = 0.047; sham-EE Mdn = 10.512 MCM2^+^ cells; IqR = 2349 vs. MCAO-EE Mdn = 14.196 MCM2^+^ cells; IqR = 5853; U = 4; *n* = 10; *p* = 0.076; sham-EE vs. sham-SD U = 1; *n* = 10; *p* = 0.016; MCAO-EE vs. MCAO-SD U = 3; *n* = 10; *p* = 0.047) (Figure 3).

### 3.4. Neural Cell Fate Choice after Stroke

To determine the impact of stroke and EE housing on adult neurogenesis, neural DCX^+^ precursors were quantified at 2 and 7 weeks post-surgery. There was a significant decrease in the number of DCX^+^ precursors in the standard group in both sham and infarct animals (2 weeks: MCAO-SD: 30.546 ± 5.875 DCX^+^ cells vs. 7 weeks: MCAO-SD: 10.030 ± 1.012 DCX^+^ cells; *p* < 0.001; 2 weeks: sham-SD: 21.318 ± 2.456 DCX^+^ cells vs. 7 weeks: sham-SD: 10.714 ± 742 DCX^+^ cells; *p* = 0.022). EE did not change the number of DCX^+^ cells in both sham and stroke animals (2 weeks: MCAO-EE: 26.507 ± 1.214 DCX^+^ cells vs. 7 weeks: MCAO-EE: 17.141 ± 840 DCX^+^ cells; *p* = 0.072; 2 weeks: sham-EE: 16.430 ± 499 DCX^+^ cells vs. 7 weeks: sham-EE: 13.117 ± 913 DCX^+^ cells; *p* = 1.000)

The number of DCX^+^ cells was not influenced by standard housing but by EE 2 weeks after infarct induction (sham-SD vs. MCAO-SD *n* = 9; *p* = 0.131; MCAO-EE vs. MCAO-SD *n* = 9; *p* = 1.000; sham-EE vs. MCAO-EE *n* = 10; *p* = 0.037; sham-EE vs. sham-SD *n* = 10; *p* = 1.000 (Figure 4)). There was also no infarct-associated increase in DCX^+^ precursor cells 7 weeks after lesion induction (MCAO-EE vs. MCAO-SD *p* = 0.513; *n* = 9) and under EE conditions (sham-EE vs. MCAO-EE *n* = 10; *p* = 1.000). No differences were observed under standard housing conditions (sham-SD vs. MCAO-SD *n* = 9; *p* = 1.000) and between the controls (sham-EE vs. sham-SD *n* = 10; *p* = 1.000 (Figure 4).

In addition, changes in proliferating DCX^+^ cells were examined 2 weeks after surgery. No significant differences between the groups could be detected (sham-SD: 73% DCX^+^MCM2^+^ cells, IqR = 12 vs. MCAO-SD: 70% DCX^+^MCM2^+^ cells, IqR = 25; *n* = 10; *p* = 0.421; sham-EE: 79% DCX^+^MCM2^+^ cells, IqR = 18.5 vs. MCAO-EE: 82% DCX^+^MCM2^+^ cells, IqR = 12.75; *n* = 8; *p* = 0.686; MCAO-EE vs. MCAO-SD *n* = 9; *p* = 0.190; sham-EE vs. sham-SD *n* = 9; *p* = 0.286 (Figure 4C,D)). 

### 3.5. Enriched Environment Increases Adult Neurogenesis but Not Aberrant Neurons after Stroke

The number of newly formed neurons showed significant changes after stroke depending on the housing conditions. Under standard housing conditions, there was a significant increase in neurogenesis in the dentate gyrus 7 weeks after a lesion (sham-SD Mdn = 1.863 EdU^+^NeuN^+^ cells; IqR = 332; MCAO-SD Mdn = 2.958 EdU^+^NeuN^+^ cells; IqR = 724; U < 0.001; *n* = 13; *p* = 0.003) (Figure 5 A,B). This increase in new neurons was detected in both the ipsi- (*p* = 0.004) and the contralateral (*p* = 0.032) side (Figure 5C). The change in housing conditions from standard to EE additionally increased neurogenesis even in the absence of a stroke lesion (*p* = 0.002). The EE significantly increased the formation of new neurons both ipsi- (*p* = 0.002) and contralaterally (*p* = 0.002) (Figure 5C). The EE animals had the highest number of neurons after the stroke. In addition, there was a significant difference in the cell numbers between sham-EE and stroke-EE groups (sham-EE Mdn = 3.827 EdU^+^NeuN^+^ cells; IqR = 544; MCAO-EE Mdn = 5.572 EdU^+^NeuN^+^ cells; IqR = 640; U = 0.00; *n* = 14; *p* = 0.002), as well as between the stroke animals due to the different housing conditions (*p* = 0.003) (Figure 5A,B). The increase in neurogenesis affected both the ipsi- (*p* = 0.009) and the contralateral (*p* = 0.002) side in the EE group (Figure 5C).

Previous studies have demonstrated the formation of aberrant neurons after stroke [30,35]. These aberrant neurons had a different morphology and an altered location in the dentate gyrus. While the new neurons in the intact brain showed a bipolar morphology with apical dendrites and a basal axon, the abnormal neurons showed bipolar dendritic trees and neurons in the hilar region [18]. These dendritic trees showed mushroom spines indicating integration into the existing network, which could be correlated with a malfunction of the hippocampal network [16,17]. A previous study in our group also demonstrated that the proportion of aberrant neurons was greatly increased by voluntary wheel running over 7 weeks after a lesion [30]. In the present study, we addressed the question of to what extent an EE and the provision of a running wheel influence aberrant neurogenesis. In this context, we show here that EE reduces the number of aberrant neurons compared with running wheel training and does not lead to any further increase compared with stroke alone. Using retrovirally labeled neurons, we found mainly neurons with bipolar morphology (Figure 5D). About 3.09% of aberrant neurons (MCAO-SD, *n* = 5/162) were detected after stroke, as compared with 8.79% after running (MCAO-running, *n* = 8/91 [30]) and 2.55% after EE (MCAO-EE, *n* = 4/157). In the control, sham-standard and sham-running animals, only two cells with atypical morphology (*n* = 2/178 (1.12%) were found. After EE alone, no aberrant neuron could be detected. Apparently, impeller training alone increases the formation of aberrant neurons, whereas the combination of multiple stimulations, as in the EE housing, reduces the generation of new aberrant neurons. 

In summary, the adult neurogenesis data show that EE with free access to running wheels significantly increases the generation of new neurons both in sham and MCAO mice. This increased neurogenesis is not accompanied by an increase in the number of aberrant neurons. In contrast, a previous study from our group showed an increase in neurogenesis after running alone, which also led to an increase in aberrant neurons [30].

### 3.6. No Changes in Morphology of Neurons due to Stroke or Enriched Housing 

Approximately 6 weeks after injection of the GFP viral vector, the majority of the newly formed neurons showed a typical morphology of granule cells (Figure 6A,B). The dendrites extended apical processes into the molecular layer (Figure 6A,B). Morphological analysis of complexity (Figure 6C,D) and dendrite length (Figure 6E) of GFP-positive neurons did not show any significant changes due to either MCAO or EE housing. 

### 3.7. Enriched Environment Housing Improves Flexible Learning in the Morris Water Maze Task

The Morris water maze task was used to assess learning performance in general and re-learning in particular. First, the platform was placed under the water surface and mice had to learn this hidden platform position within 3 days (Figure 7). All mice showed typical learning behavior, starting with the highest latency and distance on the first day. Both the latency and distance decreased steadily on the second and third day in all groups, indicating that mice learned the platform position. From day 4 on, the platform position was changed to the opposite quadrant and mice had to learn the new platform position. This re-learning was reflected in an increased latency and distance on day 4 in all groups, which then decreased on day 5. The comparison between the groups reached significance within the standard group (Figure 7A,E). Both latency and distance were significantly increased in the MCAO group compared with the sham group over all 5 days (sham-SD vs. MCAO-SD: latency: *p* = 0.042, F(1) = 5.318; distance: *p* = 0.025, F(1) = 6.607; velocity: *p* = 0.872, F(1) = 0.027).

Both latency and distance were significantly increased in the MCAO group compared with the sham group over all 5 days (sham-SD vs. MCAO-SD: latency: *p* = 0.042, F(1) = 5.318; distance: *p* = 0.025, F(1) = 6.607; velocity: *p* = 0.872, F(1) = 0.027). The direct comparison of the two groups between the individual days showed a significant increase in the distance in the MCAO group as compared with the sham group on day 1 (sham-SD vs. MCAO-SD: latency: D1: *p* = 0.068; F(1) = 4.105; D2: *p* = 0.432; F(1) = 0.664; D3: *p* = 0.328; F(1) = 1.050; D4: *p* = 0.234; F(1) = 1.586; D5: *p* = 0.088; F(1) = 3.502; distance: D1: *p* = 0.021; F(1) = 7.215; D2: *p* = 0.217; F(1) = 1.712; D3: *p* = 0.336; F(1) = 1.012; D4: *p* = 0.275; F(1) = 1.322; D5: *p* = 0.156; F(1) = 2.323; velocity: D1: *p* = 0.448; F(1) = 0.618; D2: *p* = 0.236; F(1) = 0.618; D3: *p* = 0.890; F(1) = 0.020; D4: *p* = 0.993; F(1) = 0.001; D5: *p* = 0.224; F(1) = 1.659). There was no difference in re-learning between the standard housing groups on days 4 and 5 (Figure 7A,E). Animals in the EE showed a tendency towards higher learning success than the animals in the standard cage. In the EE group, the sham, as well as the MCAO mice, showed no significant differences in the learning curves (sham-EE vs. MCAO-EE: latency: *p* = 0.283, F(1) = 1.264; distance: *p* = 0.156, F(1) = 2.294; velocity: *p* = 0.512, F(1) = 0.457) (Figure 7B/F). The direct comparison on the different days indicated significant differences on day 5 in latency and distance. The sham-EE group showed significantly better re-learning than the MCAO-EE group at day 5 (sham-EE vs. MCAO-EE: latency: D1: *p* = 0.659; F(1) = 0.205; D2: *p* = 0.302; F(1) = 1.162; D3: *p* = 0.176; F(1) = 2.066; D4: *p* = 0.636; F(1) = 0.236; D5: *p* = 0.026; F(1) = 6.429; distance: D1: *p* = 0.652; F(1) = 0.213; D2: *p* = 0.730; F(1) = 0.124; D3: *p* = 0.197; F(1) = 1.869; D4: *p* = 0.888; F(1) = 0.021; D5: *p* = 0.027; F(1) = 6.352; velocity: D1: *p* = 0.495; F(1) = 0.496; D2: *p* = 0.272; F(1) = 1.327; D3: *p* = 0.681; F(1) = 0.177; D4: *p* = 0.686; F(1) = 0.171; D5: *p* = 0.945; F(1) = 0.005) (Figure 7B–F). Previous studies [19] already showed that EE housing improves learning compared with standard housing conditions (Figure 7C,G). In our study, we found no significant changes between standard and EE housing (sham-EE vs. sham-SD: latency: *p* = 0.337, F(1) = 1.002; distance: *p* = 0.364, F(1) = 0.892; velocity: *p* = 0.13, F(1) = 8.544). Only the velocity on days 1 and 2 in the EE group showed a significant increase (sham-EE vs. sham-SD latency: D1: *p* = 0.855; F(1) = 0.035; D2: *p* = 0.796; F(1) = 0.070; D3: *p* = 0.464; F(1) = 0.572; D4: *p* = 0.190; F(1) = 1.933; D5: *p* = 0.060; F(1) = 4.304; distance: D1: *p* = 0.502; F(1) = 0.478; D2: *p* = 0.213; F(1) = 1.732; D3: *p* = 0.267; F(1) = 1.357; D4: *p* = 0.959; F(1) = 0.003; D5: *p* = 0.093; F(1) = 3.338; velocity: D1: *p* = 0.007; F(1) = 10.600; D2: *p* = 0.001; F(1) = 22.233; D3: *p* = 0.429; F(1) = 0.670; D4: *p* = 0.058; F(1) = 4.379; D5: *p* = 0.294; F(1) = 1.202) (Figure 7C,G). There were no significant differences between MCAO mice groups under the two housing conditions over the 5 days (MCAO-EE vs. MCAO-SD: latency: *p* = 0.155, F(1) = 2.331; distance: *p* = 0.416, F(1) = 0.715; velocity: *p* = 0.228, F(1) = 1.627) (Figure 7D,H). However, the comparison by day revealed significant changes in the latency and distance on day 1. 

Here, the MCAO-EE animals showed better learning performance compared with the MCAO mice in the standard cage (MCAO-EE vs. MCAO-SD latency: D1: *p* = 0.001; F(1) = 21.006; D2: *p* = 0.542; F(1) = 0.396; D3: *p* = 0.543; F(1) = 3.94; D4: *p* = 0.111; F(1) = 3.005; D5: *p* = 0.132; F(1) = 2.646; distance: D1: *p* = 0.006; F(1) = 11.662; D2: *p* = 0.528; F(1) = 0.425; D3: *p* = 0.630; F(1) = 0.245; D4: *p* = 0.517; F(1) = 0.448; D5: *p* = 0.276; F(1) = 1.312; velocity: D1: *p* = 0.143; F(1) = 2.491; D2: *p* = 0.561; F(1) = 0.359; D3: *p* = 0.726; F(1) = 0.129; D4: *p* = 0.178; F(1) = 2.067; D5: *p* = 0.103; F(1) = 3.167) (Figure 7D,H).

The probe trial took place on day 4 before the platform position was changed and on day 5 after the platform was placed on the opposite side. The probe trial served to verify the learned platform position (Figure 8). On day 4 (SE quadrant: sham-SD Mdn = 34%; MCAO-SD Mdn = 39%; U = 16.50; *n* = 13; *p* = 0.534; sham-EE Mdn = 33%; MCAO-EE Mdn = 30%; U = 13.50; *n* = 14; *p* = 0.165; sham-EE Mdn = 34%; sham-SD Mdn = 33%; U = 24.00; *n* = 14; *p* = 1.000; MCAO-EE Mdn = 39%; MCAO-SD Mdn = 30%; U = 9.50; *n* = 13; *p* = 0.101; Figure 8A) as well as on day 5 (NW quadrant: sham-SD Mdn = 35%; MCAO-SD Mdn = 29%; U = 18.00; *n* = 13; *p* = 0.731; sham-EE Mdn = 34%; MCAO-EE Mdn = 30%; U = 10.00; *n* = 14; *p* = 0.073; sham-EE vs. sham-SD; U = 24.00; *n* = 14; *p* = 1.000; MCAO-EE vs. MCAO-SD; U = 19.00; *n* = 13; *p* = 0.836; Figure 8B), animals showed an increased preference for the previously learned platform position. No differences due to MCAO or EE housing were found. 

In order to determine the quality of the platform searching, hippocampus-dependent strategies and individual strategies used were analyzed (Figure 9). In this context, the study by Woitke et al. (2017) [30] showed that the MCAO-SD animals exhibit deficits in the learning curve on days 1 and 3 compared with the sham-SD group. In addition, the MCAO animals in standard housing showed significantly lower use of the direct search strategy on day 2 and day 5. In contrast to the changes within the standard group, no differences in hippocampus-dependent learning could be detected in the learning behavior of the EE groups in the first 3 days. After switching the platform to a new position, MCAO mice in EE housing showed a lower use of hippocampus-dependent strategies (MCAO-EE vs. sham-EE group: D1 *p* = 0.399, 95% CI (0.258;1.715); D2 *p* = 0.515, 95% CI (0.206;2.210); D3 *p* = 0.099, 95% CI (0.122;1.20); D4 *p* = 0.053, 95% CI (0.161;1.013); D5 *p* = 0.041, 95% CI (0.063;0.943) (Figure 9A). Analysis of the single strategies indicates differences in the hippocampus-independent strategies on day 1 (scanning; *p* = 0.009) and day 5 (chaining; *p* = 0.031). On day 3, the sham-EE mice showed significantly higher use of hippocampus-dependent strategies. (focal search; *p* = 0.030) (Figure 9B). 

The comparison between standard and EE sham groups indicates a significantly higher use of the hippocampus-dependent strategies by the standard sham group compared with EE sham on day 3 (sham-EE vs. sham-SD hippocampal-dependent strategies D1 *p* = 0.763, 95% CI (0.248; 2.784); D2 *p* = 0.263, 95% CI (0.603; 6.400); D3 *p* = 0.010, 95% CI (1.284; 6.113); D4 *p* = 0.315, 95% CI (0.256; 1.551); D5 *p* = 0.115, 95% CI (0.073; 1.328) (Figure 9C). The analysis of the individual strategies revealed a significantly higher use of the hippocampus-dependent strategy direct search (*p* = 0.041) in the EE group on day 4 (Figure 9D). The analysis of the standard and EE MCAO groups showed significant changes in the hippocampus-dependent strategies on day 1 in favor of the EE animals (MCAO-EE vs. MCAO-SD hippocampal-dependent strategies D1 *p* = 0.047, 95% CI (0.063;0.981); D2 *p* = 0.948, 95% CI (0.311;2.983); D3 *p* = 0.582, 95% CI (0.345;6.674); D4 *p* = 0.657, 95% CI (0.242;2.445); D5 *p* = 0.169, 95% CI (0.101;1.495) (Figure 9E). The analysis of individual strategies also shows significant use of thigmotaxis (*p* = 0.040) and random search (*p* = 0.001) in the MCAO-SD group on day 1, while the MCAO-EE group searched for the hidden platform using the scanning strategy (*p* = 0.001) (Figure 9F). In the flexible learning, the MCAO-SD animals showed a significant increase in the use of hippocampus-independent strategy- scanning (*p* = 0.026) on day 5.

Taken together, the behavioral data indicate a slight improvement in learning abilities as indicated by the use of both hippocampus-dependent and -independent strategies in the EE environment groups in contrast to a previous study in our lab which showed a decline in learning ability associated with running alone [30]. Here, we present evidence indicating that the combination of an EE with running is more efficient than running alone in preserving cognitive abilities and reducing the generation of aberrant neurons after stroke.

## 4. Discussion

The present study investigated the effects of EE on adult neurogenesis and the associated generation of aberrant neurons and cognitive decline after a stroke lesion. We found that 7 weeks of EE after ischemia induction significantly increased the formation of adult-born neurons by approximately 50% compared with animals with cerebral infarction alone. However, animals in the EE showed no significant increase in the formation of aberrant neurons and no significant impairment of cognitive behavior, in contrast to animals in standard housing, which showed a post-stroke impairment. In addition, we show that the survival of DCX^+^ precursors is improved by the EE compared with the standard housing, and consequently, more precursors that differentiate into new neurons are present.

Adult neurogenesis involves complex cellular developmental stages ranging from the proliferation of stem cells to the formation of progenitor cells, their differentiation into mature neurons, and integration into the existing networks [42,43,44,45]. As part of their development, maturing neurons transiently form a basal dendrite [46]. However, this basal dendrite, which is a typical feature of aberrant cells, regresses during the maturation process [47,48]. Thus, after four to six weeks, cells with stable synaptic connections finally emerge, and the proportion of aberrant neurons with a continuing basal dendrite or a localization in the hilus is reduced to less than one percent [49]. 

Stroke considerably alters this complex sequence in adult neurogenesis [19,50]. Following a stroke lesion, stem cell and progenitor proliferation increase massively, and the differentiation of progenitor cells into mature neurons is accelerated [28,51,52]. 

The newly generated neurons also become integrated into preexisting networks [21]. In addition to the direct cellular changes, the entire neurogenic niche also reacts to the lesion. The microglia can no longer adequately remove the developing apoptotic cells, and the balance between inhibitory and excitatory excitability is altered [53,54]. Furthermore, as a consequence of multiple intrinsic and extrinsic changes in the dentate gyrus, aberrant neurons are formed [33,35,55]. This phenomenon of aberrant adult-born neurons after a lesion is well known from epilepsy studies, where the proportion of aberrant cells is even higher and also associated with a strong increase in neurogenesis through the proliferation of neural cells [36,56,57]. After experimental seizures, it was shown that these abnormally formed neurons get integrated into preexisting networks, leading to their disruption and to a deterioration in readiness and consolidation of memory content [23,36,38,55,56,58,59]. Therefore, it was postulated that aberrant neurogenesis after stroke contributes to cognitive decline afterward [30,33]. 

Besides drug treatment, rehabilitative training is the most important therapy in stroke patients [14,60]. However, it is unclear to what extent it influences aberrant neurogenesis and cognitive function in the long term. From previous studies, it is known that running wheel training mainly stimulates the proliferation of neural cells and thus increases neurogenesis [61,62]. Interestingly, this additional increase in post-stroke neurogenesis after physical training leads to an impairment in learning [21,33], while the abolition of newly generated neurons leads to an improvement [33]. Increased neurogenesis and impaired learning after stroke and voluntary wheel running were associated with an increased formation of aberrant neurons without improving learning abilities, as determined with the Morris water maze task [30]. Taken together, combined stimulation of progenitor cell proliferation by stroke and post-lesional physical activity seems to promote the formation of aberrant neurons and impair cognitive function [19]. On the contrary, maintenance in an EE does not increase the proliferation of the progenitors but rather the survival of the already-formed new cells [63,64]. These findings clearly suggest that the various activation paradigms differently influence proliferation and differentiation as well as the survival of the neural cells. 

In the present study, we found that the EE significantly potentiates the stroke-mediated increases in neurogenesis, in agreement with previous studies, but does not further promote the formation of aberrant neurons [21,29,65]. The increase in the number of newborn typical but not aberrant neurons after stroke raises the question as to what extent the formation of aberrant neurons is associated with changes in the progenitor cell population. Ablation of precursor cells prevents the formation of aberrant neurons and reduces the miswiring of the neuronal networks in the dentate gyrus, thereby decreasing seizure frequency and reducing cognitive deficits [38,66,67,68]. Therefore, we also determined the impact of standard and EE housing paradigms and investigated the changes in the neural cells and their differentiation and integration. We found that the standard housing but not the EE increased proliferation of the DCX^+^ precursors after stroke in the dentate gyrus. However, there was a significant decrease in the number of DCX^+^ precursor cells after standard housing in both the sham and stroke groups. This decrease might be explained by the reduced survival of the precursors. In this regard, Rudolph et al. already found an early increase in apoptosis within the first 10 days. In contrast, EE leads to the long-term survival of the DCX^+^ precursors with the increased formation of new neurons [54].

The altered development of the DCX^+^ precursors after stroke found here is similar to the one found in an epilepsy model, which is reinforced by the running task and contributes to increased aberrant neurogenesis [19]. In addition to the increase in neurogenesis, the integration of the newly formed neurons is crucial for their influence on the existing hippocampal network [44,45]. In a previous study, after morphological examination of newly formed neurons by transfecting dividing cells with a GFP-coupled retrovirus, we identified an aberrant neuronal morphology deviating from the physiological state in a significant proportion of the cells [35]. 

Here, we found that 3.1% of the cells fulfilled the criteria for an aberrant neuronal morphology after stroke in standard housing and 2.55% after stroke with additional EE housing. In the control group without stroke, only 1.4% of these cells were detected in the standard housing and none in the EE group. Aberrant neurogenesis has been described more extensively after epilepsy, and particularly high rates of aberrant neurons of up to 34% have been detected in rats [36]. The basis of this epilepsy-induced increase in neurogenesis is analogous to the stimulation by stroke, namely a strong increase in the proliferation of neural cells [56,59,69]. The comparable smaller amount of aberrant neurons after stroke increases when the proliferation is further stimulated by voluntary wheel running and reached 8,9% in a former study in our group [30]. This indicates a maladaptive role of enhanced physical activity after brain infarction.

The significance of aberrant neurons and the extent to which they influence network activity and disrupt cognitive behavior are still not well understood. In the field of epilepsy, a reduced propensity to seizures and epilepsy-associated cognitive deficits after a reduction in aberrant neurons was described [38,67], suggesting that cognitive outcomes might be directly correlated with the presence of aberrant neurons [55]. This raises the question of to what extent an EE reduces aberrant neurogenesis and affects cognitive function. We show that EE increases neurogenesis but not the formation of aberrant neurons, while there is no significant impairment in spatial learning, as indicated by performance in the Morris water Maze task. Mice housed under standard conditions show a significantly reduced performance as reflected by an increase in the latency to reach the platform as well as in the learning strategies. When compared with results from our former study, animals with access to a running wheel show significantly worse learning after stroke compared with animals with sham operation in relation to the latencies as well as the search strategies [30]. Taken together, a significant impairment in spatial learning after stroke is prominent in mice living in a standard cage as well as after running activity only, but not in mice living in an EE. Similar results were obtained after epilepsy. Here, housing in an EE leads to a consistent reduction of the cognitive impairments inflicted by seizures [70,71,72,73]. These findings are supported by some previous studies suggesting that combined physical and cognitive training has a positive effect on spatial-cognitive performance [29,61,74] and improves functional outcomes, both in experimental [75,76,77,78] and clinical trials after stroke [14,16]. However, further trials, including long-term follow-ups and comprehensive neurophysiological testing, are still needed to determine whether combined aerobic exercise and cognitive training lead to additive benefits following stroke [79]. In summary, our study confirms the positive impact of EE on neuronal cell morphology and cognitive function after a stroke lesion, especially as compared with physical exercise alone. It is now of particular interest to determine whether the improved learning performance is due to increased neurogenesis and what proportion of the new neurons are involved in rehabilitation. 

## 5. Conclusions

Our study shows that early placement in an EE after a stroke lesion markedly increases neurogenesis and flexible learning but not the formation of aberrant neurons, indicating that rehabilitative training as a combination of running wheel training and EE housing improves functional and structural outcomes after stroke. 

## Figures and Tables

**Figure 1 cells-12-00652-f001:**
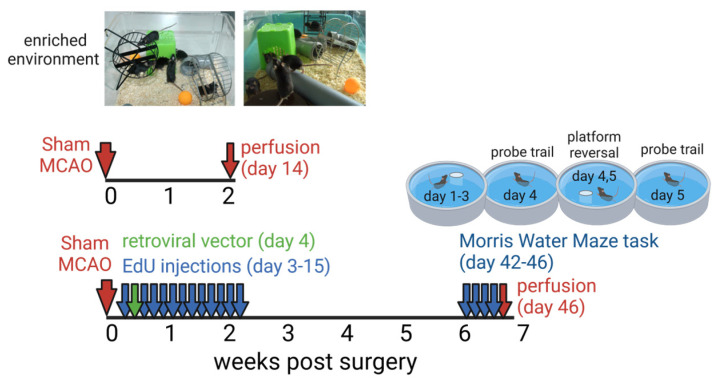
Experimental design. All animals were randomly assigned to 4 groups: 2 groups underwent a sham surgery and 2 groups an MCAO with survival period of 2 or 7 weeks. Subsequently, animals were transferred to a standard cage (Sham-SD; MCAO-SD) or to an EE housing (Sham-EE; MCAO-EE). Mice with a survival period of 7 weeks were injected with the proliferation marker EdU over a period of 3 to 15 days post-surgery and, on day 4, with the retroviral vector. At six weeks post-surgery, the spatial learning abilities were tested with the Morris water maze task over a period of 5 days. This was followed by transcardial perfusion, tissue collection and analysis. Created with BioRender.com.

**Figure 3 cells-12-00652-f003:**
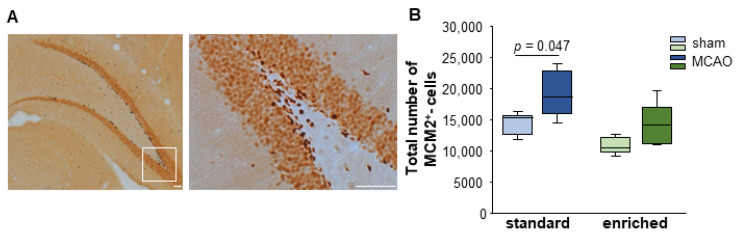
Proliferation of MCM2^+^ cells after stroke and EE housing. (**A**) Peroxidase-stained images of MCM2^+^ cells in the dentate gyrus. Dark brownish cells were identified as MCM2^+^ cells. (**B**) Significant changes in the endogenous proliferation in the dentate gyrus were detected under standard housing conditions. Statistical analysis was performed using the Mann–Whitney U test, scale bar = 100 µm, higher magnification 10 µm.

**Figure 4 cells-12-00652-f004:**
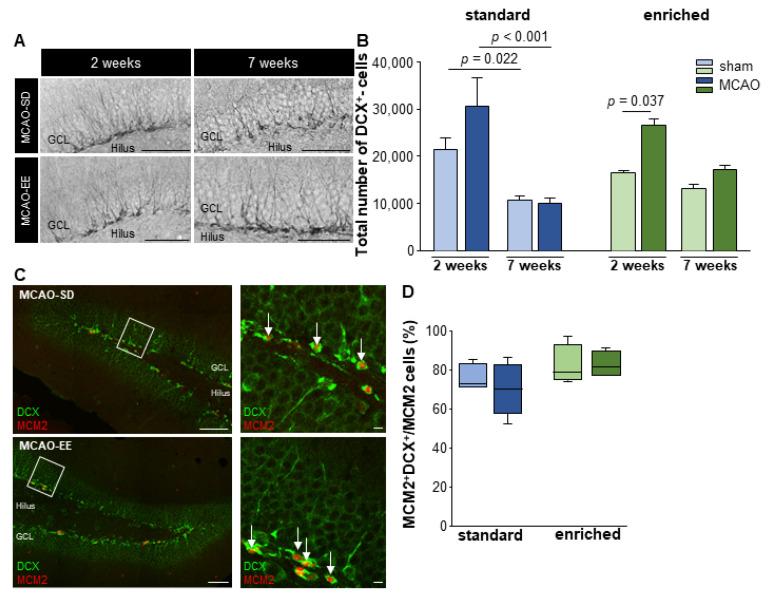
Impact of stroke and EE on DCX^+^ cell proliferation. (**A**) Peroxidase-stained brain slices for DCX 2 and 7 weeks after stroke. (**B**) Quantification of DCX^+^ cells in the dentate gyrus. The absolute number of DCX^+^ cells significantly decreased after stroke in the standard but not in the EE groups. Statistical analysis was performed using the ANOVA and post-hoc Bonferroni, MW ± SEM; scale bar = 100 µm. (**C**) Exemplary immunofluorescence staining for MCM2 (red) and DCX (green) after MCAO in standard and EE housing groups. (**D**) Percentage of proliferating DCX^+^ cells from all proliferating cells in the dentate gyrus. Statistical analysis was performed using the Mann–Whitney U test, scale bar = 100 µm, higher magnification 10 µm.

**Figure 5 cells-12-00652-f005:**
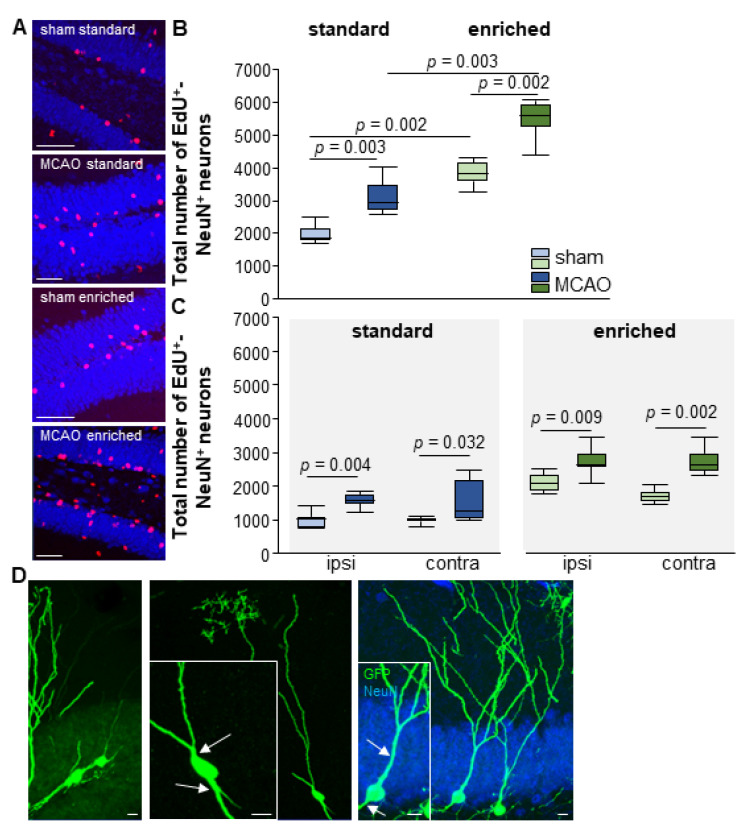
Stroke and EE-induced neurogenesis. (**A**,**B**) MCAO induction significantly increased new neuron formation 7 weeks later. EE significantly increases the number of newly generated neurons in both sham and MCAO mice. (**C**) This increase occured ipsi- and contralaterally in all groups. The boxplots show the median, upper and lower quartile, and minimum and maximum values. Statistical analysis was performed using the Mann-Whitney U test, scale bar = 50 µm. (**D**) Aberrant GFP-labelled new neurons with two dendritic trees (white arrows), one dendritic tree extending into the hilus (GFP green, NeuN blue). Scale bar = 10 µm. For the morphological analyses of the newly born neurons, we determined the following numbers of cells: MCAO-SD *n* = 20; MCAO-sham *n* = 28; MCAO-EE *n* = 24; sham-EE *n* = 9. Scale bar = 10 µm.

**Figure 6 cells-12-00652-f006:**
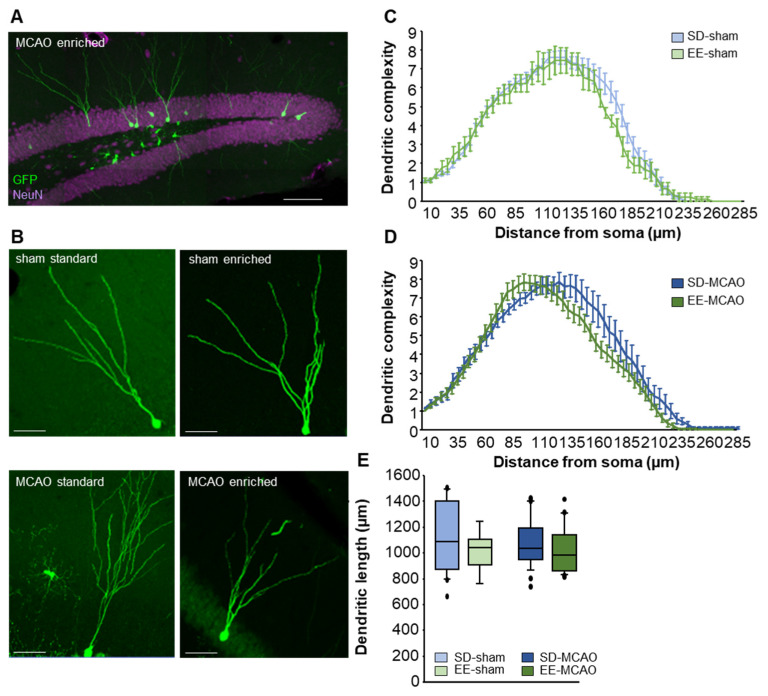
Morphological analysis of newly generated neurons in the dentate gyrus. (**A**) Overview images GFP-labelled neurons in the dentate gyrus of an EE-MCAO animal (GFP green, NeuN purple). Scale bar = 100 µm (**B**) Representative confocal images of 42 days GFP-positive neurons in the different groups post-surgery. Scale bar = 50 µm. (**C**–**E**) Dendritic length and complexity of newly generated GFP-positive neurons in the different groups. No significant differences were found between sham and MCAO groups. Statistical analysis was performed using linear mixed model for dendritic length and One-way ANOVA with post-hoc Bonferroni for dendritic complexity.

**Figure 7 cells-12-00652-f007:**
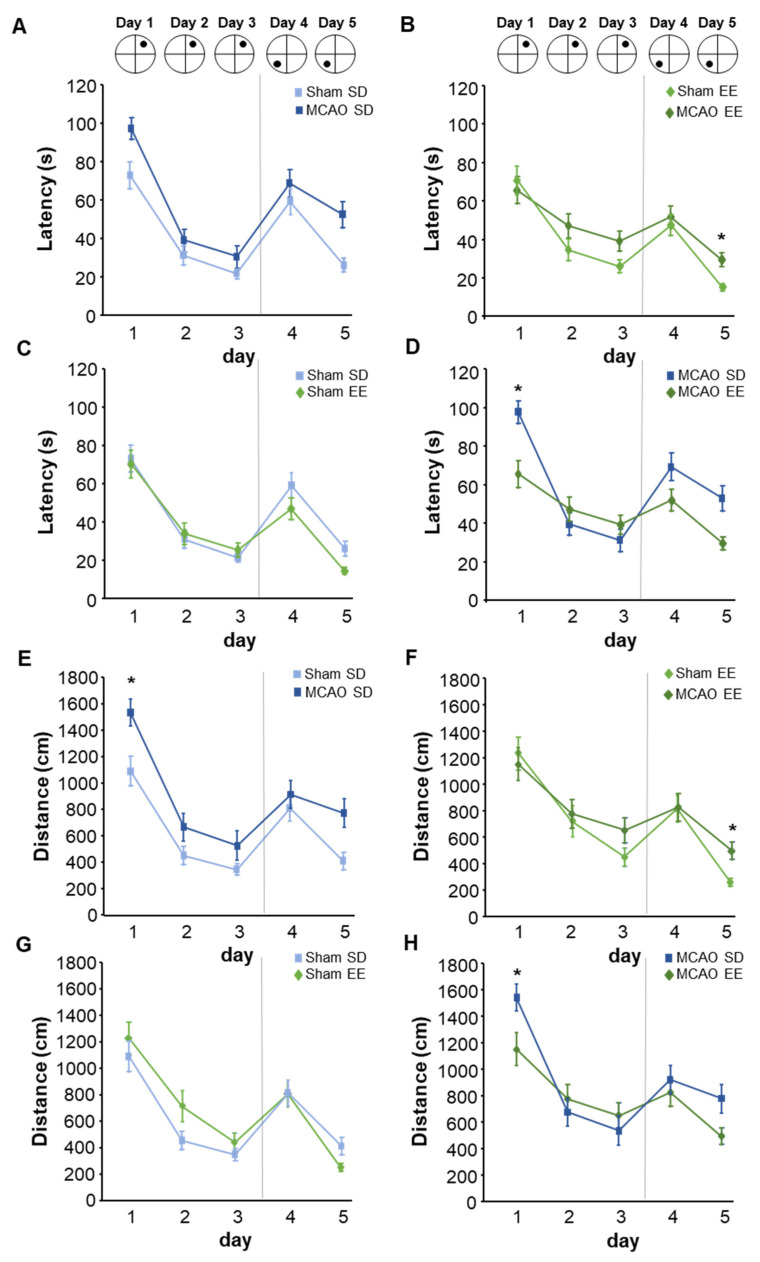
Learning and re-learning in dependence of stroke and EE housing. The graphs show (**A**–**D**) latency and (**E**–**H**) distance of navigation to the hidden platform in the Morris water maze. (**A**,**E**) Impairments in learning were evident in the standard housing group when comparing sham and MCAO. (**B**,**F**) By re-learning the new platform position, sham-EE animals, in particular, showed an improvement compared with the MCAO-EE mice. (**C**,**G**) There were no differences between the sham groups. (**D**,**H**) Between the MCAO groups in the different housing conditions, there was a significant impairment in learning on day 1 in the MCAO-SD group. Graphs show mean ± SEM using 2-way ANOVA with repeated measurements and post-hoc Bonferroni (per group and per day), * *p* < 0.05.

**Figure 8 cells-12-00652-f008:**
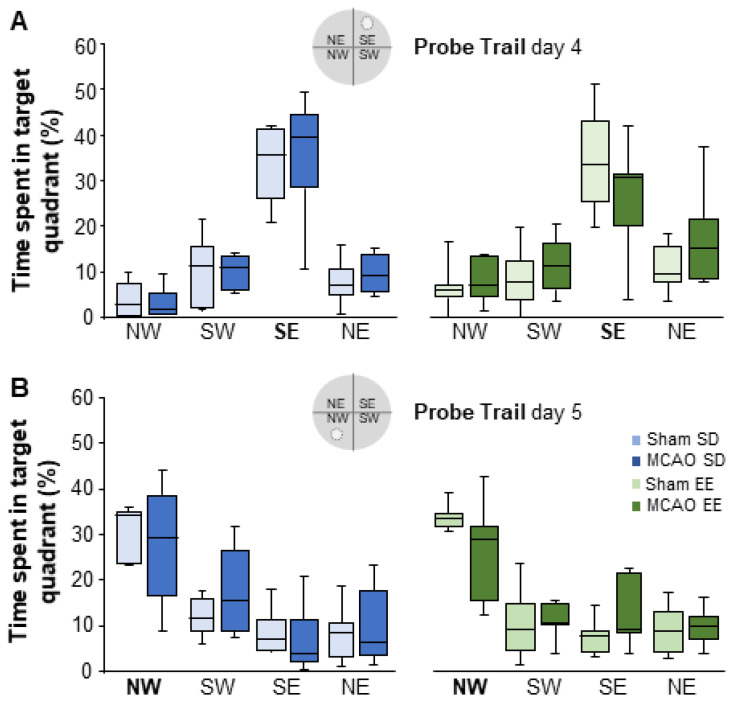
The trial was conducted on (**A**) day 4 (before the platform change) and (**B**) day 5 (after the platform change). The platform was removed from the pool for the probe trail, and the time spent in the different quadrants was determined. (**A**) All animals learned the first platform position in the quadrant SE without significant differences between the groups. (**B**) On day 5, all groups showed a higher preference for the new platform position. There was no impairment in spatial learning during the probe trial. The boxplots show the median, the upper and lower quartiles and the minimum and maximum values. The analysis was performed with the Mann-Whitney U test, all *p*- and *n*-values. SD = standard housing; EE = enriched environment housing.

**Figure 9 cells-12-00652-f009:**
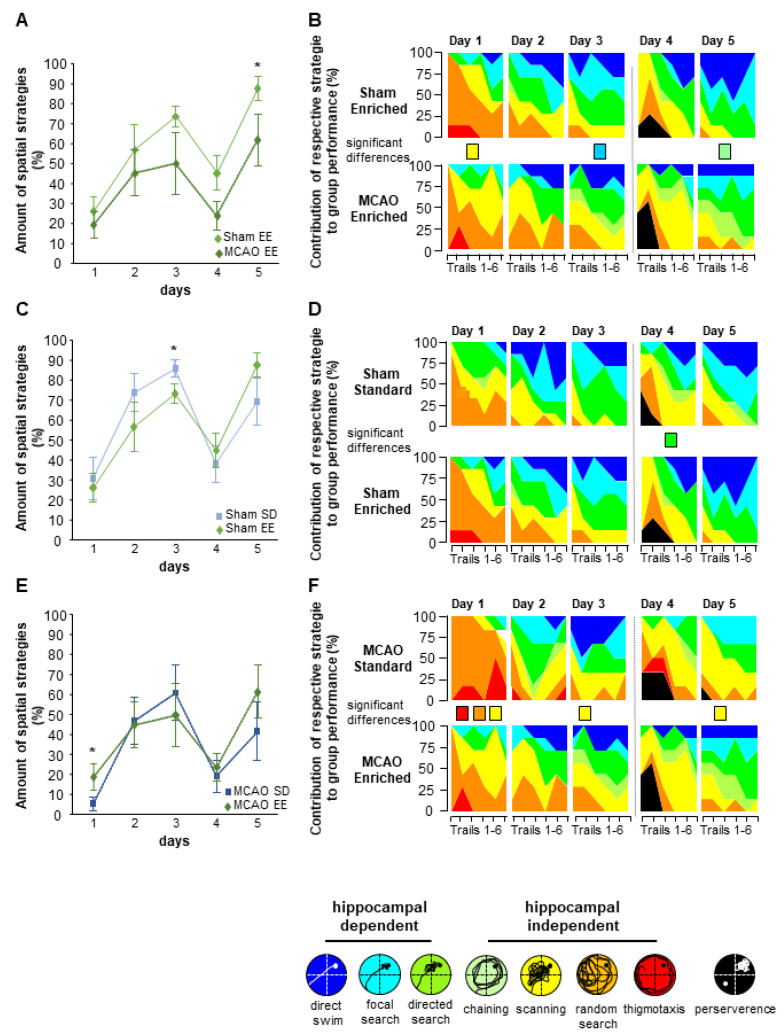
Analysis of hippocampus-dependent and independent search strategies in the MWM test. (**A**,**C**,**E**) Representation of hippocampus-dependent strategies over time. The data show an improved learning behavior in the EE sham group compared with the EE MCAO. Furthermore, there was more frequent use of the hippocampus-dependent strategies on day 1 in the MCAO-EE group compared with the MCAO-SD group. (**B**) Within the hippocampus-independent strategies, the MCAO-SD animals showed increased use of the scanning strategy on day 5. For analysis of the different hippocampal-dependent and -independent search strategies on each day, an exploratory data analysis by means of an algorithm based on the generalized estimating equations method was performed. Significant differences in strategies usage are represented by colorful squares for each day, * *p* < 0.05.

## Data Availability

The data presented in this study are available in the article.

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
