# Peer review of "Post-Stroke Environmental Enrichment Improves Neurogenesis and Cognitive Function and Reduces the Generation of Aberrant Neurons in the Mouse Hippocampus"

_cells, 2023, doi:10.3390/cells12040652_

Round 1

Reviewer 1 Report (Previous Reviewer 1)

Cell 2023 Revision

Woitke et al.

Post-stroke environmental enrichment improves neurogenesis and cognitive function and reduces the generation of aberrant neurons in the mouse hippocampus

MCAO induces the generation of new neurons in the hippocampus. However, due to their aberrant morphology these neurons may not contribute to improved cognition. In their revised study, Woitke et al. determined whether ENR might diminish structural cell changes and consequently improve behavior outcome in mice at 46 days post-stroke. The authors compared 4 groups (sham SD, sham EE, MCAO SD, MCAO EE), measured cell numbers (EDU/NeuN), cell morphology (rv-GFP), and determined spatial memory by Morris water maze. They added cell proliferation data 2 weeks post-stroke (MCM2+ cells) – yet, I am surprised that the known huge increase in cell proliferation was not observed in MCAO SD – and MCM2/DCX overlap. The story is still very enriched by text and statistics. The English writing, and organization (reduction) of the discussion has much improved. Nevertheless, I am still wondering whether only 2.5 to 3.0% of aberrant neuron so strongly contribute to the behavior outcome – and given the authors comparison to 30% in epilepsy stories. Please see more specific comments below:

1.     2.6. line 219, and 2.8. line 286, the authors added MCM2 staining to get the absolute number of proliferating cells; however, when stating absolute numbers, at least every 6th to 8th slice should be counted! I believe, every 24th was used because staining was done on slices left? The authors could count EDU+ cells instead, in addition?! (same for DCX, every 24th is too little for absolute numbers, the multiplying factor is too high)

2.     3.3, line 480 (and 219, throughout), please carefully distinguish progenitors/precursor cells; while DCX-expressing cells indeed describe an intermediate phase of the neuronal lineage, cell proliferation (such as in RUN) increases precursor cells that might not become neurons, and are rather referred to as neural cells. The headline should be e.g. “Neuronal fate choice after stroke”. When describing/talking about DCX cells, e.g. line 536, 538, 564 and throughout), please write “DCX-positive (or -expressing) cells” instead of ‘neuronal precursor”. Line 540, it should be written: “… enriched housing groups.” (plural)! For the results, graph Fig. 4B ENR, it does look like MCAO decreases DCX numbers at & weeks, while it is already much less compared with MCAO SD? Line 547, the sentences is not finished and “Fig. 5” is missing in the text; Fig. 4 and 5 could actually be put together

3.     Please update Fig.3 to Fig. 6 in the text! Starting line 568, and follow-up through all figures till Fig. 9

4.     3.5. Fig. 8 (Fig. 5 before; the Fig. numbers should be updated within the text!! In addition, the *** are missing throughout Fig. 8), the authors did not acknowledge my comment: when comparing groups, the experimental group is compared to CTL, e.g. sham EE to sham SD, MCAO EE to SD EE, and not vice versa. It should state: (e.g. line 800, “B/F … latency and distance were significantly increased in MCAO-EE mice compared with sham-EE. (D/H) Between MCAO groups, latency and distance were significantly lower at day 1 and day 5 for the EE group compared to MACO SD. Here in addition, a ‘decline’ cannot be shown since its only day 1, and no comparisons to an earlier time point for the same group exists.  

5.     Abbreviations – the authors use EE throughout Figures and Legends, it might help to be consistent throughout the text also

6.     English writing, e.g. line 992, instead of ‘did not affect’, ‘had no effect on xxx’;

7.     Line 994/5, Run has been shown to only increase cell proliferation short-term, and has not effect long-term, consequently, DCX cells are reduced, yet, the NET effect on cell numbers is not decreased (compared to standard housing)!

8.     Line 1008,Disc: ”.. after stroke IN standard housing conditions, and 2.55 % after stroke …”

Author Response

  1. 6. line 219, and 2.8. line 286, the authors added MCM2 staining to get the absolute number of proliferating cells; however, when stating absolute numbers, at least every 6th to 8th slice should be counted! I believe, every 24th was used because staining was done on slices left? The authors could count EDU+ cells instead, in addition?! (same for DCX, every 24th is too little for absolute numbers, the multiplying factor is too high)

Response: We followed the reviewer's recommendation and quantified again the MCM2 and DCX-positive cells from each 6th section using peroxidase staining. Results show that the stroke in the standard group but not in the enriched group increases proliferation 2 weeks post-infarct.

The updated DCX counting confirms the previous trend and shows a significant increase in DCX cells in the enriched MCAO versus enriched sham group.

  1. 3.3, line 480 (and 219, throughout), please carefully distinguish progenitors/precursor cells; while DCX-expressing cells indeed describe an intermediate phase of the neuronal lineage, cell proliferation (such as in RUN) increases precursor cells that might not become neurons, and are rather referred to as neural cells. The headline should be e.g. “Neuronal fate choice after stroke”. When describing/talking about DCX cells, e.g. line 536, 538, 564 and throughout), please write “DCX-positive (or -expressing) cells” instead of ‘neuronal precursor”. Line 540, it should be written: “… enriched housing groups.” (plural)! For the results, graph Fig. 4B ENR, it does look like MCAO decreases DCX numbers at & weeks, while it is already much less compared with MCAO SD? Line 547, the sentences is not finished and “Fig. 5” is missing in the text; Fig. 4 and 5 could actually be put together

Response: We followed the reviewer’s suggestion and improved the text accordingly. The headline was changed to “Neuronal cell fate choice after stroke” (line 486). “DCX-positive cells” has been included instead of “neuronal precursors” where appropriate. The missing plural has been updated (line 520). Results in Fig. 4B: this comparison was not significantly different. The incomplete sentence was partially covered by the underlying figure (line 527). This has been corrected. Fig. 4 and 5 have been put together, as recommended by the reviewer.

  1. Please update Fig.3 to Fig. 6 in the text! Starting line 568, and follow-up through all figures till Fig. 9

Response: We apologize for the wrong notation. All figures have now been updated in the text.

  1. 3.5. Fig. 8 (Fig. 5 before; the Fig. numbers should be updated within the text!! In addition, the *** are missing throughout Fig. 8), the authors did not acknowledge my comment: when comparing groups, the experimental group is compared to CTL, e.g. sham EE to sham SD, MCAO EE to SD EE, and not vice versa. It should state: (e.g. line 800, “B/F … latency and distance were significantly increased in MCAO-EE mice compared with sham-EE. (D/H) Between MCAO groups, latency and distance were significantly lower at day 1 and day 5 for the EE group compared to MACO SD. Here in addition, a ‘decline’ cannot be shown since its only day 1, and no comparisons to an earlier time point for the same group exists.

Response: We adjusted the text when there was a direct comparison between SD and EE and listed EE first.

Thank you for the recommendation, we have replaced “decline” with “impairment”.

  1. Abbreviations – the authors use EE throughout Figures and Legends, it might help to be consistent throughout the text also

Response: We now included the abbreviation “EE” for “enriched environment” throughout the text, including the figure legends.

  1. English writing, e.g. line 992, instead of ‘did not affect’, ‘had no effect on xxx’;

Response: The sentence has been changed accordingly (line 988).

  1. Line 994/5, Run has been shown to only increase cell proliferation short-term, and has not effect long-term, consequently, DCX cells are reduced, yet, the NET effect on cell numbers is not decreased (compared to standard housing)!

Response: We agree with the reviewer that we cannot speculate on the effects of running on proliferation and precursor cells. We therefore focus on standard and enriched housing to describe the changes in proliferation and DCX+ cells. The extent to which running affects the DCX+ precursors remain an open question.

  1. Line 1008,Disc: ”.. after stroke IN standard housing conditions, and 2.55 % after stroke …”

Response: The sentence has been improved accordingly (line 1003).

Reviewer 2 Report (New Reviewer)

The manuscript ‘Post-stroke environmental enrichment improves neurogenesis and cognitive function and reduces the generation of aberrant neurons in the mouse hippocampus ‘by Florus Woitke et.al utilized the mice MCAO which represents the stroke model in rodents and investigated how enriched environment affects the formation of aberrant neurons and cognitive function after a stroke lesion. Authors used EdU for proliferating cell labeling and retroviral vector for aberrant neurons detection. In addition, they used Morris Water Maze (MWM) which is a classic rodent behavior test for cognitive function especially learning and memory. Authors designed the experiments and found that enriched environment significantly increases neurogenesis and flexible learning after MCAO, while did not increase the number of aberrant neurons in the dentate gyrus. The study indicates that rehabilitative training as a combination of running wheel training and enriched environment housing improves functional and structural outcomes after stroke. Overall, the manuscript is well written, the methods part and the results parts were clearly presented with figures. In the discussion part, authors also cited enough reference for adult neurogenesis and how increased neurogenesis especially post-stroke neurogenesis will lead to impairment in learning. Authors found that enriched environment significantly potentiates the stroke-mediated increases in neurogenesis and indicates that the combination strategy with running wheel training improves the outcomes after stroke. I think the manuscript is acceptable for publication on Cells.

Author Response

no further comments

This manuscript is a resubmission of an earlier submission. The following is a list of the peer review reports and author responses from that submission.

Round 1

Reviewer 1 Report

Cells 2022

Woitke et al.

Post-stroke environmental enrichment improves neurogenesis and cognitive function and reduces the generation of aberrant neurons in the mouse hippocampus

Ischemic stroke often leads to personal and socioeconomic consequences, and is among the 10 leading causes of death. Induced cerebral ischemia (e.g. MCAo) in rodent brain is used to define post-stroke acute and long-term structural, anatomical alterations and changes in animal behavior. Ischemia robustly increases adult neurogenesis in the hippocampus-which plays a central role in learning and the encoding and retrieval of spatial memories. Novel experience such as physical exercise or exposure to an enriched environment (ENR) also enhance cell proliferation and survival. Yet, the MCAo-induced new neurons often display aberrant morphology and do not improve cognitive function. In the current study, Woitke et al. studied whether post-stroke ENR might enhance the structural and behavior outcome in mice 46 days post-stroke. The authors compared 4 groups and measured newly generated cell numbers (EDU/NeuN) and morphology (rv-GFP) in hippocampus, and determined spatial memory by Morris water maze. In comparison to their previous publication, Run together w ENR enhances AD and thus also improves cognition. The story is very detailed and quite interesting and the results support the idea of involved neuron generation in the process. However, some data should be improved in their visibility while text could be reduced. Please see my specific questions and comments below.

Results:

Did authors confirm stroke severity by behavior tests such as RotaRod or PoleTest?

3.3. / 4., Please write absolut numbers into the text (not Mdn or IqR)

GFP(NeuN) and dendrites, please provide an overview image of the DG, how many cells were analyzed? All mentioned comparison w/ old/already published results belong to the discussion; please provide new data/graphic here. If authors argue that MCAo EE group has 50% more new neurons, yet no further increase in the amount of aberrant cells, how is this reflected in GFP(NeuN)-expressing cells? How do authors explain that ‘only’ 3% are aberrant, and would this small amount affect behavior (w/ 50% more new neurons generated)?

3.5., the manuscript is very detailed; yet for water maze results, it’s too elaborate and focus on main findings perish: comparison of results by conditions before and post-stroke are of most importance (sham SD-EE, and MCAo SD-EE; is sham EE vs MCAo EE relevant for behavior analysis?). Please also review the description of results, e.g. Fig. 5 legend, L490/1, although sham EE animals show improved learning behavior, the focus should be on EE post-stroke and written the other way around (such as ‘.. mice in MCAo EE did not improved in learning as much as sham EE..’; the same for D/H: yet, EE is improved over SD).

Please think of using abbreviations also in the text (EE, SD etc), and keep the one in use throughout (e.g. ABN)

The discussion is also very detailed and should be shortened/condensed; e.g. l580 to 660 is repetitive, and relevant data discussion here is only in l616 .., and l650 to 660. Overall, a few sentences on the relevance for clinical improvement should be provided.

Reviewer 2 Report

The authors detected the potential effects of enriched environment on dentate neurogenesis with MCAO mice. They found that enriched environment obviously increases newly generated neurons with normal morphology, while decreases the aberrant neurons, which may be related to the improved performance in Morris water maze task. I don’t think the manuscript is suitable to be published at present form since some major issues have not been clarified.

  1. EdU was used to label the dividing cells in the dentate gyrus and was examined at 7 week after MCAO. However, the proliferate rate of progenitor cells in the SGZ, the survival ability of newly born cells as well as their neuronal differentiation selection definitely contributed to the results observed by the authors. More detail experiments should be designed and performed to determine which one lead to an increase of mature neurons.
  2. A causal relationship between increase neurogenesis/reduced aberrant neurons and improved cognitive function has not been established in this study.
  3. I am not sure whether the model used in this study has made a significant lesion in the hippocampus, especially in the dentate gyrus because the authors did not show the infarct range regarding to the dentate gyrus.
  4. When cell counting was done, I am not sure whether the target cells were counted in the all areas of DG, including granule cells layer, molecular layer and hilus?
  5. Please show the aberrant neurons using intact dentate gyrus imagine.

Round 2

Reviewer 2 Report

I don't believe the manuscript has been sufficiently improved to warrant publication in Cells.

Author Response

Reviewer 2: EdU was used to label the dividing cells in the dentate gyrus and was examined at 7 weeks after MCAO. However, the proliferate rate of progenitor cells in the SGZ, the survival ability of newly born cells as well as their neuronal differentiation selection definitely contributed to the results observed by the authors. More detail experiments should be designed and performed to determine which one lead to an increase of mature neurons.

We have addressed the criticisms regarding the lack of proliferation after stroke and enriched environment and have conducted further experiments.

For the experiments, we allowed the animals to survive under the different conditions for a period of 2 weeks and evaluated the total number of proliferating cells, the number of proliferating DCX+ neuronal progenitors and the total number of neuronal DCX+ progenitors. With the new results, we can further show that keeping the animals in the EE does not increase the proliferation but the survival of the DCX neuronal progenitors, which correlates with an increased number of new neurons in the dentate gyrus. These results are consistent with previous literature indicating that EE-associated maintenance is responsible for cell survival and not proliferation.

Enclosed please find the revised manuscript entitled „Post-stroke environmental enrichment improves neurogenesis and cognitive function and reduces the generation of aberrant neurons in the mouse hippocampus” by Florus Woitke, Antonia Blank, Anna-Lena Fleischer, Shanshan Zhang, Gina-Marie Lehmann, Julius Broesske, Madlen Haase, Christoph Redecker, Christian W. Schmeer and Silke Keiner, which we want to resubmit for publication as a Regular Manuscript in Cells.
